# Burden and factors associated with ongoing transmission of soil-transmitted helminths infections among the adult population: A community-based cross-sectional survey in Muleba district, Tanzania

**Franco Zacharia**[1], **Valeria Silvestri**[1]*, **Vivian Mushi**[1,2], **George Ogweno**[1], **Twilumba Makene**[1], **Lwidiko E. Mhamilawa**[1]

**1** Department of Parasitology and Medical Entomology, Muhimbili University of Health and Allied Sciences, Dar es Salaam, Tanzania, **2** Department of Zoology and Wildlife Conservation, College of Natural and Applied Sciences, University of Dar es Salaam, Dar es Salaam, Tanzania

* silvestri.valeria82@gmail.com

## Abstract

### Background

In Tanzania, school-based Mass Drug Administration (MDA) campaigns have been the main strategy for the prevention and control of Soil Transmitted Helminths (STH) infection. Adults are not part of the program and could remain as the reservoir of infection, favoring continuity in transmission. Water, Sanitation, and Hygiene (WaSH) issues and slow progress in community awareness promotion campaigns contribute to the persistence of STH as public health issue among target populations notwithstanding the achievements of the control interventions.

### Objective

This study aimed to determine the current prevalence and the risk factors associated with ongoing transmission of STH infection among adults in Muleba District, Tanzania.

### Methodology

A household-based quantitative cross-sectional study was carried out among 552 adults in Muleba district. Through a quantitative interviewer-administered questionnaire, information was registered related to socio-demographic characteristics, level of knowledge on the disease, and WaSH factors. The prevalence of STH and estimation of its intensity were assessed by analyzing stool samples through formol-ether concentration and the Kato-Katz technique. Descriptive statistics was used to summarise data; logistic regression to determine the association between STH infection and socio-demographic and WaSH factors. A p-value < 0.05 was considered statistically significant.

**Data Availability Statement:** All relevant data are within the paper and its Supporting Information files.

**Funding:** The author(s) received no specific funding for this work.

**Competing interests:** The authors have declared that no competing interests exist.

## Results

A total of 552 adults were included in the study; 50.7% (280/552) were female. The median age was of 30 years, ranging from 18 to 73 years. A prevalence of 9.1% (50/552) for STH infection was reported; the prevalence of *Hookworm Spp.*, *Ascaris lumbricoides*, and *Trichuris trichiura* was 7.43%, 0.91%, and 0.72%, respectively. The factors significantly associated with STH infection were farming (aOR = 3.34, 95% CI: 1.45–7.70), the habit of not wearing shoes in general (aOR = 5.11, 95% CI: 1.55–16.87), and during garden activities (aOR = 4.89, 95% CI: 1.47–16.28).

## Conclusions and recommendations

We observed an aggregated prevalence of STH infections (*Ancylostoma duodenale*, *Trichuris trichiura*, and *Ascaris lumbricoides*) of 9.1% among the adult population, indicating a decreasing prevalence but ongoing transmission. Integrated management is needed to address practices contributing to ongoing transmission.

## Introduction

Soil-transmitted helminths (STH) are among the Neglected Tropical Diseases of gastroenterological medical importance for humans and include the roundworm (*Ascaris lumbricoides*), the whipworm (*Trichuris trichiura*), hookworms (*Necator americanus* and *Ancylostoma duodenale*) and the thread-worm (*Strongyloides stercoralis*). While the infectious route of *Trichuris trichiura* and *Ascaris lumbricoides* is oral-fecal, through ingestion of infective eggs, Hookworm invasion occurs through skin penetration of an infective larval stage of the parasite, while *S. stercoralis* can infect humans both orally, through auto-infection of larvae from intestinal eggs, and percutaneously [1]. The infection leads to nutritional impairment, iron loss, and anemia; morbidity and mortality are associated with the intensity of infection and age and immunity of the host [1, 2]. Globally, STH affect about 1.5 billion people, with an estimated burden above 3 million disability-adjusted life years (DALY) [3]. Endemicity is worldwide and prevalence is higher in areas with poor access to safe water sources and sanitation among populations with low hygienic standards, including low-income and middle-income countries [1].

Children are among the vulnerable population with 568 million school-age children living globally in high risk areas [1]. Education level, occupation, hand washing habits, latrine usage, and contact with soil have been previously acknowledged as additional factors contributing to infection [2]. Adults are not exempt from getting infected. Different previous studies from other settings (Ethiopia and Ecuador) have analyzed the prevalence of STH infection among the adult population ranging from 31.2% to 65% with a co-infection rate ranging from 0.8% to 25% [4, 5]. According to WHO data, worldwide, 844 million people (58% living in sub-Saharan Africa) have no access to basic drinking water service while 2.3 billion people still lack access to fundamental sanitary facilities [6], thus favoring exposure and re-infection with STH in absence of WaSH services [7, 8].

Additionally, knowledge attitudes and practices related to STH infection among adults are still poor, with diffused open defecation practices (ODP), or habit of walking without shoes or slippers, which favors re-infections of STH after treatment [9–11]. The WHO strategic plan 2011–2029 for the prevention and control of STH infection included the provision of anti-worm drugs programs (Albendazole 400 mg once a year for a prevalence of STH between 20

and 50%, or twice a year for a prevalence higher than 50%) focusing on populations at risk, which includes kids, women of reproductive age (15–49 years) and pregnant women (second and third trimester) [12]). Because MDA does not prevent the re-infection, an integrated strategy that includes a safe water supply and health education for behavior change is desirable [13, 14].

In Tanzania, previous studies on geohelminths and *Schistosoma* prevalence conducted in North West Tanzania have observed an overall prevalence of 6.7% [15]. Data from Pemba, in the Zanzibar archipelago, confirmed that STH infections are still endemic, despite control measures in place since the 1990s [16], with an overall prevalence of STH up to 85.4% [17]. In Tanzania, the Participatory Hygiene and Sanitation Transformation programme (PHAST) improved the water supply, and led to a steady decline in STH infections, especially from 2009 to 2012 [14]. Still, studies are needed to better understand factors that can favor infection among the adult population, which is excluded by the MDA program.

Our study aimed to investigate the current prevalence and risk factors associated with ongoing transmission of STH infection (specifically WaSH factors and knowledge, attitudes and practice [KAP] factors) among adults in Muleba District, Tanzania. Findings could further guide stakeholders towards an integration of the prevention and control programmes that are actually in place with activities targeting adults, towards achieving the WHO 2030 Global targets for STH.

## Materials and methods

### Study setting

Muleba is among the six districts of the Kagera Region, located on the western shore of Lake Victoria, in the northern part of Tanzania. Muleba is composed of five divisions with 32 wards and has an area of 3444 $Km^2$, with 7925 Km of Lake Victoria water body containing 20 islands. Fishing and agriculture, pastoralism, and small-scale mining are their main economic activities. According to the 2022 Tanzania National Census, the population of the Muleba District is estimated to be 637,659 people (315073 male and 322586 female) [18]. Rainfall occurs in two seasons: the "short rains" in October–December (average monthly rainfall 160 mm) and the "long rains" in March–May (average monthly rainfall 300 mm) [19]. Muleba was purposely selected as the setting for this study because it is among the districts in Tanzania with a history of a high prevalence of STH [20–22].

### Study design

A community-based cross-sectional study conducted at the household level was designed to determine the current prevalence of STH and risk factors associated with ongoing transmission among the adult population in this setting. Data collection was carried out from April to May 2022.

### Study population, inclusion, and exclusion criteria

Adult participants aged 18 and above, residents in the Muleba district for at least 3 months (time required from infection to detection of eggs in stool) were considered for enrollment. A recent (≤3 months) history of anthelminthic treatment, and mental impairment were considered criteria for exclusion.

The sample size was obtained by the use of the formula [23]: n = $Z^2$p [1-p]/$\varepsilon^2$ whereby; n = the minimum estimated sample size; Z = standard normal deviate (1.96 for 95% confidence interval); p = expected proportion, corresponding to 32% from previous studies in the

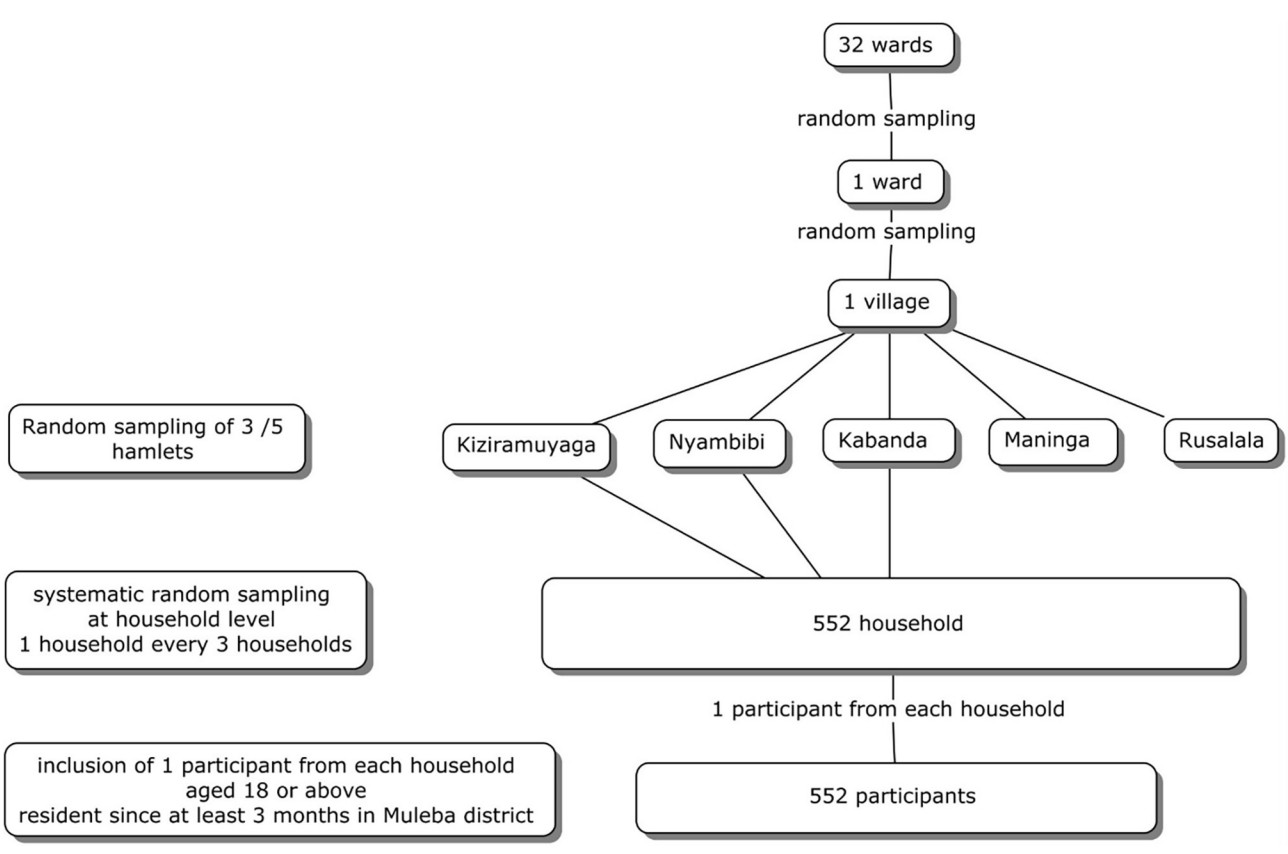

**Fig 1. Sampling of participants.** Flow chart.

Lake Victoria region [24], ε = the margin of error settled at 5%; a design effect of 1.5 and considering 10% of anticipated non-response rate. A sample size of 552 participants was estimated.

The sampling procedure was conducted using a three-stage cluster sampling method: in the 1st stage, a simple random sampling of one ward out of 32 was performed, followed in the second stage by a simple random sampling of one village among the ones in the selected ward. In the third stage, household sample units were randomly selected in the village using the lottery method, to reach a total number of 552. From each household, one participant was enrolled according to inclusion criteria. The oldest in the household was chosen in cases where more than one eligible participant was available. A flowchart of the sampling process is provided [Fig 1].

## Data collection tools and processes

**Questionnaire.** An interviewer-administered questionnaire with closed-ended questions, developed in English and translated into Kiswahili was used to obtain information on the socio-demographic characteristics of participants, knowledge, and attitudes towards STH infection, and availability and usage of WaSH factors. Participants knowledge related to STH infection (modality of infection, preventive measures) was explored through three questions, concerning the previous acquisition of information on STH infections, participants perceptions on whether walking with barefoot and open defecation can facilitate STH infection or if wearing shoes, using toilets and washing hands with soap before preparation or consumption

of food and after toilet visits can prevent people from acquiring STH infection. WaSH factors related to exposure (availability and accessibility of water and its usage, nature and distance of the water source from household, status of sanitation including type of toilets, presence of hand washing facilities, conditions of personal hygiene) were analyzed. Finally, a section that inquired on attitudes towards toilet use and practices (hand washing and shoes wearing, toilet use at home or in the field, hand and food washing before eating) was also added. Pre-testing of the questionnaire was carried out among 30 adults randomly selected in the Bukoba district in Kagera. The participants involved in the testing were not included in the study. The questionnaire was then administered to participants in each household, by trained community health personnel.

**Laboratory investigations.** Pre-labeled stool containers were given to every participant, instructed to provide at least 10 g of the specimen. At acceptance, samples were preserved using formalin (10%) for transport to MUHAS laboratory. Stool samples were consecutively processed using the formal-ether concentration technique and analyzed with optical microscopy assessment by experienced laboratory technicians [25]. Positive samples were then further analysed to quantify the intensity of the infection by using the Kato- Katz egg counting technique. The stool samples were processed to make one single Kato-Katz thick smear covered with cellophane soaked in glycerine and malachite green [26]. Samples were then analysed through optical microscopy, to quantify the intensity of infection by one microscopist.

**Quality assessment.** Data collection tools were pre-tested and the research assistants were trained. Standard operating procedures (SOP) were followed during specimen collection, transportation, processing, examination, and result recording under the supervision of the researcher. Quality assessment of the diagnostic process was carried out by cross-checking 10% of the available samples by a second investigator.

## Statistical analysis

Statistical analysis was carried out using the STATA Corp software version 14.0 (STATA Corp, College Station, TX, USA). Independent and dependent variables were summarized using descriptive statistics, reported as mean and standard deviation for continuous variables and frequencies and proportions for categorical ones. A $\chi^2$ was used to compare categorical variables. The intensity of STH infection was measured by counting the number of eggs per gram of stool sample collected; the intensity level was classified as defined by WHO guidelines; thresholds for moderate and heavy infections were 5000 and 50,000 EPG for *A. lumbricoides*, 1000 and 10,000 EPG for *T. trichiura*, and 2000 and 4000 EPG for hookworm, respectively [27].

The knowledge questions were scored 1 for a right answer and 0 for a wrong one. A total score of 0–1 was regarded as "poor knowledge," while a total score of 2–3 was regarded as "good knowledge."A cut-off p-value of 0.2 was used to select variables to include in the logistic regression analysis, to assess the strength of the association between the independent and dependent variables. The association was expressed through Crude Odds Ratio (COR) and Adjusted Odds Ratio (AOR) after adjusting for confounders. A p-value of less than 0.05 was considered statistically significant.

## Ethical approval

The project received approval from the Muhimbili University of Health and Allied Sciences institutional review board (MUH-REC-05-2022-1137). Informed consent was acquired in written form; given the non-invasive nature of the investigations, the adult population, and the absence of participants with intellect impairment, verbal consent was considered for those not-able to write. Confidentiality was ensured. Permission was acquired from the local

authorities, specifically by the District Executive Officer of the Muleba district. Privacy and confidentiality were ensured. Patients found positive for STH infection were referred to the health care facility for treatment. The procedures followed were in accordance with the ethical standards of the Helsinki Declaration (1964, amended most recently in 2008) of the World Medical Association.

## Results

### Socio-demographic characteristics of the study participants

A total of 552 adults living in Kiziramuyaga village were included in the study. About 50.7% (280/552) were women. The median age was 30 years (15 years in the interquartile range), ranging from 18 to 73 years. According to marital status, 53.8% were married and as for literacy, 55.8% had a primary level of education. The most represented occupation was farming (67.7%). Anagraphic data are summarized in Table 1.

### Prevalence and intensity of STH among the study participants

The observed prevalence of STH was 9.1% (50/552). According to species, 7.43% (41/552) were hookworms, 0.91% (5/552) were *A. lumbricoides* and 0.72% (4/552) were *T. trichiura*. No mixed infections were observed.

**Table 1. Anagraphic factors and STH infection.**

| Variable | N (%) | Infected (N = 50) n/N (%) | cOR | p-value <0.05 | aOR | p-value <0.05 |
|---|---|---|---|---|---|---|
| **Sex** | | | | | | |
| Male | 272/552 (49.3) | 22/272 (8.1) | Ref | 0.435 | | |
| Female | 280/552 (50.7) | 28/280(10.0) | 1.26 (0.70–2.27) | | | |
| **Age group** | | | | | | |
| <25 | 140/552 (25.4) | 13/140 (9.3) | Ref | | Ref | |
| 25–34 | 204/552 (36.9) | 27/204(13.2) | 1.49 (0.74–3.00) | 0.264 | 4.56 (0.44–47.25) | 0.203 |
| 35–44 | 115/552 (20.8) | 6/115 (5.2) | 0.54 (0.20–1.46) | 0.224 | 7.23 (0.81–64.72) | 0.077 |
| 45–54 | 47/552 (8.5) | 3/47 (6.4) | 0.67 (0.18–2.45) | 0.541 | 2.30 (0.24–22.01) | 0.469 |
| <54 | 46/552 (8.3) | 1/46 (2.2) | 0.22 (0.03–1.71) | 0.147 | 2.65 (0.25–28.00) | 0.418 |
| **Marital status** | | | | | | |
| Married | 297/552 (53.8) | 24/297 (8.1) | Ref | | Ref | |
| Single | 236/552 (42.7) | 24/236(10.2) | 0.75 (0.16–3.42) | 0.708 | 0.58 (0.11–2.90) | 0.505 |
| Other | 19/552 (3.4) | 2/19 (10.5) | 0.96 (0.21–4.42) | 0.961 | 0.74 (0.13–4.14) | 0.730 |
| **Level of education** | | | | | | |
| None | 42/552 (7.6) | 3/42 (7.1) | Ref | | | |
| Primary | 308/552 (55.8) | 29/308 (9.4) | 1.35 (0.39–4.65) | 0.633 | | |
| Secondary | 184/552 (33.3) | 18/184 (9.8) | 1.41 (0.40–5.02) | 0.597 | | |
| Collage/University | 18/552 (3.3) | 0/18 (0.0) | - | - | | |
| **Occupation** | | | | | | |
| Farmers | 374/552 (67.8) | 43/374(11.5) | 3.17(1.39–7.20) | 0.006* | 3.34 (1.45–7.70) | 0.005* |
| Other | 178/552 (32.2) | 7/178 (3.9) | Ref | | Ref | |
| **Family size** | | | | | | |
| <3 | 95/552 (17.2) | 91/95 (95.8) | Ref | | Ref | |
| 3–4 | 292/552 (52.9) | 260/292 (89) | 1.18 (0.21–6.70) | 0.846 | 0.57 (0.88—.75) | 0.561 |
| 5–6 | 109/552 (19.7) | 97/109 (89.0) | 3.32 (0.77–14.22) | 0.107 | 1.7 (0.35–8.60) | 0.501 |
| >6 | 56/552 (10.1) | 54/56 (96.4) | 3.34 (0.72–15.5) | 0.123 | 2.21 (0.43–11.3) | 0.342 |

cOR Stands for Crude Odds Ratios, aOR Stands for Adjusted Odds Ratios, *Statistical significance at p<0.05

Participants aged 25–34 years had a prevalence of 13.2%, but no statistical difference was observed for prevalence between different age groups (cOR = 1.49, 95% CI: 0.74–3.00; p = 0.264 and aOR = 4.56, 95% CI = 0.44–47.25; p = 0.2). Additionally, prevalence did not significantly differ according to participants' gender, marriage status, size of family nor education level. When analyzing STH prevalence in relation to occupation, it was significantly higher among farmers (43/374; 11.5%) compared to other workers (7/178; 3.9%), with a cOR of 3.17 (95% CI = 1.39–7.20; p = 0.006, confirmed by an aOR of 3.34(95% CI = 1.45–7.70; p = 0.005 (Table 1).

## Water supply and prevalence of STH among the study participants

Unprotected spring was reported to be the main source of water for domestic use (346/552; 62.7%), followed by river (132/552; 23.9%), with no significant difference in prevalence of infection among participants fetching water from different water sources. No other significant difference in prevalence was reported according to water consumption or the distance to the water source [Table 2].

## Sanitation, hygiene factors, and STH infection among the study participants

When analyzing sanitation and hygiene related factors, the presence of a toilet was not associated with the prevalence of STH infection, but a significant association was initially found according to the toilet meeting the required standards (prevalence of 37/497; 7.4% for participants with toilets meeting the standard vs 13/55; 23.6% for participants with toilet not meeting standards, with a cOR = 3.84, 95% CI = 1.90–7.80 [Table 3]. The presence of hand washing facilities was also initially negatively associated with reduced prevalence of infection compared

**Table 2. Water factors and STH infection.**

| Factor | N (%) | Infected (%) | p-value ($\chi^2$) |
|---|---|---|---|
| **Source of water** | | | |
| River | 132/552 (23.9) | 14/132 (10.6) | 0.477 |
| Unprotected spring | 346/552 (62.7) | 30/346 (8.7) | 0.681 |
| Pond | 4/552 (0.7) | 0/4 (0.0) | 0.526 |
| Unprotected well | 2/552 (0.4) | 0/2 (0.0) | 0.655 |
| Rainwater | 2/552 (0.4) | 0/2 (0.0) | 0.655 |
| Protected spring | 3/552 (0.5) | 1/3 (33.3) | 0.142 |
| Protect well | 2/552 (0.4) | 0/2 (0.0) | 0.655 |
| Pipe | 75/552 (13.6) | 5/75 (6.7) | 0.438 |
| Other | 2/552 (0.4) | 0/2 (0.0) | 0.655 |
| **How much water do the family members consume per day** | | | |
| 20L | 3/552 (0.5%) | 0/3 (0.0) | |
| 21-40L | 14/552 (2.5%) | 1/14 (7.1) | |
| 41-60L | 97/552 (17.6%) | 7/97 (7.2) | |
| 61-80L | 320/552 (57.9%) | 32/320 (10) | |
| >80 | 118/552 (21.4%) | 10/118 (8.5) | 0.886 |
| **For how long do you travel to get water for domestic use?** | | | |
| <30 | 190/552 (34.4%) | 13/190 (6.8) | |
| 30–120 | 338/552 (61.2%) | 35/338 (10.4) | |
| 121–240 | 22/552 (3.9%) | 2/22 (9.1) | |
| >240 | 2/552 (0.4%) | 0/2 (0.0) | 0.568 |

**Table 3. Sanitation factors and STH infection.**

| Factor | N (%) | STH (%) | | cOR | P-value | aOR | p-value <0.05 |
|---|---|---|---|---|---|---|---|
| Does the family have toilet? | | | | | | | |
| Yes | 548/552(99.3) | 49/248(8.9) | | 0.295 | 0.294 | | |
| No | 04/552(0.7) | 1/4(25.0) | | (0.03–2.89) Ref | | | |
| Types of Toilets | | | | | | | |
| Safely managed | 130/552(23.55) | 9/130(6.9) | | Ref | | Ref | |
| Basic | 379/552(68.6) | 38/379(10.0) | | 1.50 (0.70–3.19) | 0.294 | 1.03 (0.47–2.29) | 0.941 |
| Shared | 38/552(6.9) | 3/38(7.9) | | 1.15 (0.30–4.50) | 0.838 | 0.80 (0.20–3.23) | 0.750 |
| Unimproved | 5/552 (0.9%) | 0/5(0.0) | | - | - | | - |
| Does the toilet meet the required standards | | | | | | | |
| Yes | 497/552(90.0) | 37/497(7.4) | | Ref | | | |
| No | 55/552(10) | 13/55(23.6) | | 3.84(1.90–7.80) | 0.001* | 2.21(0.97–5.03) | 0.059 |
| Presence of Hand washing Facilities | | | | | | | |
| Yes | 326/552(59.1) | 20/326(6.1) | | Ref | | | |
| No | 226/552(40.9) | 30/226(13.2) | | 2.34(1.29–4.23) | 0.005* | 1.03(0.35–3.04) | 0.958 |
| Availability of Water and Soap | | | | | | | |
| Yes | 125/552(22.6) | 4/125(3.2) | * | Ref | | | |
| No | 427/552(77.3) | 46/427(10.7) | | 3.65(1.28–10.35) | 0.015* | 2.55(0.8–8.06) | 0.112 |
| Location of Hand washing Facilities | | | | | | | |
| Yes | 318/552(57.6) | 21/318(6.6) | | Ref | | | |
| No | 234/552(42.4) | 29/234(12.4) | | 2.00(1.11–3.60) | 0.021* | 1.36(0.5–3.68) | 0.551 |
| Signs of usage (toilets & hand washing facilities) | | | | | | | |
| Yes | 529/552(95.8) | 45/529(8.5) | | Ref | | | |
| No | 23/552(4.2) | 5/23(21.7) | | 2.98(1.05–8.43) | 0.039* | 1.42(0.44–4.54) | 0.558 |

cOR Stands for Crude Odds Ratios, aOR Stands for Adjusted Odds Ratios, *Statistical significance at p<0.05

to the absence of these facilities, with a prevalence respectively of 20/326; 6.1% vs 30/226; 13.2% among participants that respectively reported and not reported the presence of washing facilities, with an increased risk among the latter and cOR of 2.34, 95% CI = 1.29–4.23; p = 0.005. The same was observed for the availability of water and soap, and participants that reported to have soap and water had a prevalence of STH infection of 4/125; 3.2% vs 46/427; 10.7% among those that didn't have these items, which had an increased risk of STH infection with a cOR of 3.65,95% CI = 1.28–10.35; p = 0.015. The location of a hand washing facility was initially associated with a lower prevalence of infection among participants (21/318; 6.6% vs 29/234 (12.4%) with an increased risk of infection for those having no washing facility, with a cOR of 2.00, 95% CI = 1.11–3.60; p = 0.021. The same for the presence of signs of usage on hand and washing facilities (prevalence of 45/529; 8.5% vs 5/23; 21.7% among those that didn't have signs of usage). The absence of signs of usage was initially associated with higher risk of infection with cOR = 2.9895% CI = 1.05–8.43; p = 0.039. None of the associations observed in univariate logistic regression analysis was confirmed by multivariate logistic regression analysis, conducted to exclude the effect of confounders.

## Knowledge and STH infections among the study participants

The majority of the study respondents (495/552; 80.6%) had poor knowledge of STH infection (total score between 0 and 1). No association was observed between knowledge score and STH infection in logistic regression analysis, nor for each of the questions of the knowledge section,

**Table 4. Knowledge factors and STH.**

| Factor | N (%) | N STH (%) | p-value | cOR | P | aOR | p-value |
|---|---|---|---|---|---|---|---|
| **Knowledge score** | | | | | | | |
| Low | 445/552 | 40/445 (10.1) | 0.01 | | | | |
| High | 57/552 | 0/57 (0) | | | | | |
| **Have you ever heard of STH infection** | | | | | | | |
| Yes | 42/552 (7.6) | 0/42 (0.0) | | | | | |
| No | 510/552 (92.4) | 50/510 (9.8) | 0.033 | | | | |
| **If yes when did you hear it?** | | | | | | | |
| 2019 | 3/552 (7.1) | 0/3 (0.0) | | | | | |
| 2020 | 32/552 (76.1) | 0/32 (0.0) | | | | | |
| 2021 | 07 (16.6) | 0/7 (0.0) | NA | | | | |
| Total | 42 (100) | 0/42 (0.0) | | | | | |
| **Are you aware that walking with bare foot and defecating can facilitate STH infection?** | | | | | | | |
| Yes | 49/552 (8.9) | 0/49 (0.0) | | | | | |
| No | 32/552 (5.8) | 3/32 (9.4) | | | | | |
| I don't know | 471/552 (85.3) | 47/471 (10.0) | 0.068 | | | | |
| **Wearing shoes, using toilet and washing hands can prevent you from STH infection?** | | | | | | | |
| Yes | 33/552 (5.9) | 0/33 (0.0) | | | | | |
| No | 44/552 (7.9) | 2/44 (4.6) | | | | | |
| I don't know | 475/552(86) | 48/475 (10.1) | 0.082 | | | | |

including if participants had heard previously about STH infection and if yes in what year; if they were aware that walking bare foot and defecating could predispose to infection and the acknowledgement of shoes and hand washing as means for prevention [Table 4].

## Attitudes and practices among the study participants

The aggregated scores obtained from responses to questions on practices and attitudes (negative attitude = below average; positive attitude = above the mean score) showed that the majority of participants had a positive attitude and reported positive practices towards STH infection (530/552; 96%). Among those that washed hands, the prevalence was lower than the one reported among those that did not comply to this practice (17/281; 6.1% vs 33/271; 12.1%), with a cOR of 2.15, 95% CI = 1.16–3.96; p = 0.014] but this finding was not confirmed after adjusting for confounding factors.

The habit of wearing shoes corresponded to a lower prevalence of infection when compared to participants not wearing them (44/538; 8.2% vs 6/14; 42.9%); an increased risk for infection among those not wearing feet protection was observed, with cOR of 8.42,95% CI:02.79–25.36, p = 0.001, confirmed on multivariate logistic analysis by an aOR of 5.11,95% CI = 1.55–16.87; p = 0.007. Similarly, participants wearing shoes or gumboots during the garden activities had a lower prevalence of infection compared to those who didn't wear protection (3/148; 25 vs 47/404; 11.6%) with a cOR of 6.36, 95% CI = 1.95–20.77; p = 0.002 confirmed by an aOR of 4.89, 95% CI = 1.47–16.28; p = 0.010) [Table 5].

## Factors associated with ongoing transmission of STH infections among the study participants

The multivariate logistic regression analysis confirmed the association with STH, after adjustment for confounders, for the farmer working activity, with an aOR of 3.34,95% CI = 1.45–

**Table 5. Practices and STH.**

| Factor | N (%) | N STH (%) | P<0.05 | cOR | p-value | aOR | p-value |
|---|---|---|---|---|---|---|---|
| **Do you wear shoes/gumboot during your garden activities** | | | | | | | |
| Yes | 148/552 (26.8%) | 3/148 (2.0) | | Ref | | | |
| No | 404/552 (73.1%) | 47/404 (11.6) | 0.001* | 6.36(1.95–20.77) | 0.002 | 4.89(1.47–16.28) | 0.010* |
| **Do you wear slippers at home** | | | | | | | |
| Yes | 543/552 (98.3%) | 50/543 (9.2) | | | | | |
| No | 9/552 (1.6%) | 0/9 (0.0) | 0.340 | | | | |
| **Do you keep your fingernail short and clean** | | | | | | | |
| Yes | 545/552 (98.7%) | 50/545 (9.2) | | | | | |
| No | 7/552 (1.26%) | 0/7 (0.0) | 0.401 | | | | |
| **Observed hand washing process** | | | | | | | |
| Yes | 281/552 (50.9) | 17/281 (6.1) | 0.012* | Ref | | | |
| No | 271/552 (41.1) | 33/271 (12.2) | | 2.15(1.16–3.96) | 0.014* | 0.97(0.33–2.84) | 0.96 |
| **Do you have the habit of wearing shoes** | | | | | | | |
| Yes | 538/552 (97.5) | 44/538 (8.2) | 0.001* | Ref | | | |
| No | 14/552(2.5) | 6/14 (42.9) | | 8.42(2.79–25.36) | 0.001* | 5.11(1.55–16.87) | 0.007* |
| **Do you wash hands after cleaning children bottom?** | | | | | | | |
| Yes | 66/552 (60.0) | 0/66 (0.0) | 0.001* | | | | |
| No | 44/552 (40) | 8/44 (18.2) | | | | | |
| **Do you defecate in toilet when at home** | | | | | | | |
| Yes | 539(97.6) | 49/539 (9.1) | 0.862 | | | | |
| No | 13 (2.4) | 1/13 (7.7) | | | | | |
| **Defecate in toilet when are in their daily activities** | | | | | | | |
| Yes | 438/552 (79.4) | 40/438(9.1) | 0.905 | | | | |
| No | 114/552 (20.6) | 10/114 (8.8) | | | | | |
| **Do you wash hands after toilet visit** | | | | | | | |
| Yes | 353/552 (63.9) | 28/353 (7.9) | 0.220 | | | | |
| No | 199/552 (36.1) | 22/199 (11.1) | | | | | |
| **Do you wash hands with soap after toilet** | | | | | | | |
| Yes | 272/552 (49.3) | 24/272 (8.8) | 0.850 | | | | |
| No | 280/552 (50.7) | 26/280 (9.3) | | | | | |
| **Do you wash your hands before eating** | | | | | | | |
| Yes | 533/552 (96.6) | 50/533 (9.4) | 0.162 | | | | |
| No | 19/552 (3.7%) | 0/19 (0.0) | | | | | |
| **Do you wash hands with soap before eating** | | | | | | | |
| Yes | 213/552(38.6) | 18/213(8.4) | 0.694 | | | | |
| No | 339/552 (61.4) | 32/339 (9.4) | | | | | |

cOR Stands for Crude Odds Ratios, aOR Stands for Adjusted Odds Ratios, *Statistical significance at p<0.05

7.70; p = 0.005, for the habit of not using shoes as feet protection, with an aOR of 5.11,95% CI = 1.55–16.87; p = 0.007 and for the one of not wearing shoes or gumboots while working in the garden, as confirmed by an aOR of 4.89,95% CI = 1.47–16.28; p = 0.010).[Tables 1, 5].

## Discussion

In the last years the MDA programs and the other interventions in place in Tanzania, including improvement of the water supply, have led to a steady decline in STH infections, especially from 2009 to 2012 [14]. As in the case of other studies on Neglected tropical diseases that are

endemic in the country, like onchocerciasis [28] and schistosomiasis [29], that assessed the prevalence and associated factors after years of MDA distribution, this study was conducted in response to the need of acquiring new evidences on the prevalence (assessed on stool samples) and factors associated with an increased risk of STH infection (exploring anagraphic, WASH related, knowledge and practice factors) in Muleba district, a region previously known to be endemic for these helminthiasis,. This to provide new data after years of interventions in place (that for STH consist of yearly MDA distribution with Albendazole among SAC, as for national guidelines).

We have found an overall prevalence for STH of 9.1% (50/552) among adult participants in the Muleba district; the prevalence included 7.43% (41/552) of *Hookworm* infections. No coinfection was reported and infection among positive participants was prevalently of light intensity. The occupational status as a farmer, not wearing shoes or gumboots during garden activities, and not wearing shoes in general were associated with an increased risk for STH infection. In a study by Siza et al. published in 2015, conducted among adults in the Lake Victoria basin to assess the prevalence of *Schistosomes* and STH and morbidity associated with schistosomiasis, a prevalence of 21.7% for hookworms, 8.3% for *Ascaris lumbricoides*, and 2.0% for *Trichuris trichiura* was reported [30]. The 9.2% prevalence observed seven years later in our study, lower than what was reported in the Lake Victoria's setting previously, suggests the effectiveness of interventions in place which include, in Tanzania, annual MDA campaigns conducted in primary schools as part of the national Neglected Tropical Diseases (NTD) control programme [21].

Differently from what reported in other settings, which showed 0.8% of coinfections with *Ascaris lumbricoides* and *Hookworm* [4], no co-infection was observed among our participants. Still, it is important to be aware of the fact that co-infection can occur due to the immunosuppressive features that can follow helminths infestation, environmental and climatic factors, and the absence of appropriate WASH infrastructure and may contribute to the detrimental proliferation of more than one helminth condition [31] leading to severe clinical manifestations [32]. As for other parasitic conditions, STH prevalence has been reported previously to decrease with older age [33], but in our study, this aspect was not statistically significant. Engagement in activities that increase contact with infested water or soil [33] and low immunity at a younger age may contribute to more susceptibility to infection in paediatric population [34], while continuous exposure to infection could favor a gradual decline in worm burdens, as a partial immunity to new infections develops in adulthood [35].

The intensity of the infection among positive participants was light in all cases as per the WHO classification criteria [27], in accordance with national data [36], and with data from other East African countries, such as Ethiopia [4]. Although prevalence is the main key metric in many STH epidemiological studies, the intensity of infection is an important determinant of the morbidity induced by STH infection. The infection's intensity is also more reliable marker of interventions success, given the non-linear relationship between prevalence and intensity (defined by the negative binomial distribution of parasite numbers per person). At low average worm loads, the prevalence falls rapidly and becomes less informative on the epidemiology of persistent transmission [12]. The low intensity observed in our setting can be interpreted as an indicator of the success of the interventions in place like the yearly MDA in primary schools as for the above mentioned national control programme for NTDs [21]. Because adults aren't targeted for receiving MDA, notwithstanding the significant prevalence observed among them, the extension of MDA at the community level could achieve an additional reduction in prevalence, which has been quantified in previous studies up to 90%, making it an attractive option for the future [37].

The current study has revealed that most of the respondents still use unprotected open water sources, even though the source of water used and the treatment of drinking water were not significantly associated with STH infection. This scenario on the quality of the water source used is in accordance with the Global report, which denounces that 263 million people spend over 30 minutes per round trip to collect water from an improved source, 159 million collect drinking water directly from surface water sources, 58% live in sub-Saharan Africa [6]. Similarly to what was observed for water-sources and treatment, the logistic regression analysis didn't confirm an association of STH infection with sanitation factors. Other factors, prevalently behaviour factors, like not wearing shoes and practicing activities at risk like, for example, farm work, play a major role in infection.

Despite the high coverage of sanitation facilities, open defecation is still practiced within the study area; 2.4% of participants don't defecate in a toilet when they are at home, and 20.6% don't use a toilet when carrying out their daily activities. Additionally, 19.4% of the stool sample tested positive for *E.coli*, which is an indicator of water contamination with fecal matter, that increases the risk of coinfection with intestinal parasites that share an oral-fecal route of transmission. This fact emphasizes the need to establish and sustain safe water sources for both drinking and washing in endemic areas [32]. Environment contamination together with other factors that predisposed to exposure, such as the occupational status as a farmer, not wearing shoes or gumboots during garden activities or not wearing shoes in general increased risk for STH infection, and are likely major drivers for infection in this setting. The environment supportiveness of STH biology, favored by ecological differences in temperature, rainfall, and vegetation (indicators of soil humidity and shade) [17, 30, 38], contributes to the persistence of transmission after soil contamination.

Even though our finding didn't reach statistical significance when analyzing the association between water sources and sanitation factors, other authors have observed that inadequate sanitation can increase the odds of infection with skin-penetrating STH species, while unimproved water supply increases the odds of infection with orally-ingested STH species [39, 40]. This observation urgently calls for WaSH interventions that could help communities to secure access to adequate quantities of water for daily needs [41]. Data from a systematic review and meta-analysis inquiring on the association between WaSH access and practices and STH infection, confirmed that WaSH access and practices are associated with lower odds of STH infection [42]. WaSH affects the STH disease burden, by reducing exposure to STH infective stages in the environment [7]. Recent data from a survey conducted in Pemba Island, Zanzibar, proved that notwithstanding the rounds of massive drug administration with albendazole and praziquantel implemented for over 25 years, targeting both children and adults, as for Zanzibar Elimination of Schistosomiasis Elimination (ZEST) Programme, the control of STH morbidity was still insufficient and poor sanitation could contribute to persistent transmission, in addition to high population density and potential resistance to treatment [17]. In contexts with a high baseline prevalence of STH, the objectives set by WHO for 2030 with the only use of PC intervention will likely not be feasible, if not accompanied by a substantial improvement of sanitation [17, 37].

The majority of participants had poor knowledge regarding the transmission and prevention of STH infection, and only a minority had high knowledge, even though no association was found between specific knowledge related variables and STH infection. Gaps in KAP were acknowledged in several studies [43, 44]. If implemented, health hygiene educational intervention and increased knowledge can improve health hygiene behavior and reduce intensity of STH [45]. As observed in other studies, knowledge needs an environment that supports its translation into practice [29]: in the lack of a supportive environment exposure will occur

notwithstanding the acknowledgement of risk, and knowledge will fail to determine a reduction of prevalence.

Finally, the majority (96%) of the participant from the current study showed positive attitudes; still, no significant difference was observed in terms of STH prevalence among participants with positive compared to negative attitudes (respectively 8.7% vs 18.2%; p = 0.128). In the absence of infrastructures and WASH facilities, positive attitudes can't be translated into positive practices [46], emphasizing the need of integrating educational programs with the building of a practice-supporting environment.

## Limitations of the study

Our study provides updated data on this important topic of epidemiological and public health concern, for which only a few previous studies are available in Tanzania. The consistent number of participants enrolled and the variety of variables assessed constitute the strength of our work. Still, some limits have to be acknowledged. The use of a questionnaire for the collection of data related to WaSH factors, infection and prevention knowledge, attitudes, and practices can be prone to recall bias. Additionally, we collected and processed only a single stool sample from each participant, one only slide was prepared from each sample and analysed by one microscopist. Even though 10% of samples were cross-checked for quality assessment, this is sub-optimal and could underestimate the prevalence of STH among the studied population.

## Conclusions and recommendations

Our study has documented the ongoing transmission of STH infection among the adult population in the Muleba district, Tanzania, showing a prevalence of 9.1% among participants older than 18 years, notwithstanding the interventions in place including MDA. The association of practices like not wearing shoes in general or not wearing shoes during specific activities prone to exposure, independently from the level of knowledge or attitudes towards infection prevention, or WASH factors, emphasized the need to increase the adhesion to the use of protective garments among the exposed population. Additionally, it is urgent to build a supportive environment to favor the translation of knowledge and attitudes in effective preventive practices. The extension of MDA to adult population, the integrated approaches of educational interventions together with the building of infrastructure, and the extension of WASH coverage will help to meet the 2030 global targets for soil-transmitted helminthiases.

## Supporting information

**S1 Data.**
(XLS)

## Author Contributions

**Conceptualization:** Franco Zacharia, Twilumba Makene, Lwidiko E. Mhamilawa.

**Data curation:** Franco Zacharia, Lwidiko E. Mhamilawa.

**Formal analysis:** Franco Zacharia.

**Investigation:** Franco Zacharia, George Ogweno.

**Methodology:** Franco Zacharia, George Ogweno, Lwidiko E. Mhamilawa.

**Project administration:** Franco Zacharia.

**Supervision:** Twilumba Makene, Lwidiko E. Mhamilawa.

**Validation:** Twilumba Makene, Lwidiko E. Mhamilawa.

**Writing – original draft:** Franco Zacharia, Valeria Silvestri, Lwidiko E. Mhamilawa.

**Writing – review & editing:** Franco Zacharia, Valeria Silvestri, Vivian Mushi, Lwidiko E. Mhamilawa.

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
