## [Decision Letter · Decision Letter 0]

11 May 2023

PONE-D-23-02659Burden and factors associated with ongoing transmission of soil-transmitted helminths infections among the adult population: a community-based cross-sectional survey in Muleba district, TanzaniaPLOS ONE

Dear Dr. Valeria,

Thank you for submitting your manuscript to PLOS ONE. After careful consideration, we feel that it has merit but does not fully meet PLOS ONE’s publication criteria as it currently stands. Therefore, we invite you to submit a revised version of the manuscript that addresses the points raised during the review process. In addition to addressing comments raised by the reviewer, kindly attend to the following observations for improving the quality of the manuscript.

Introduction:

The authors mention STH without mentioning Strongyloides stercoralis as one important member of the STH group of helminths.

**Methods**

*Study settings*

The authors have used 2012 population for Muleba while there is a new report with a different population. Kindly refer to the 2022 National Census report https://www.nbs.go.tz/nbs/takwimu/Census2022/Administrative_units_Population_Distribution_Report_Tanzania_volume1a.pdf

*Statistical analysis*

Line 186: An independent sample t-test or χ2 was used to compare continuous and categorical variables. This statement is not clear, which one of the two was compared using t-test and which one with χ2 Please, make it clear.It is also worthwhile noting that, conducting the statistical tests intends at estab lishing whether there is an association between the independent and the dependent variable. The strength of logistic regression analyisis is that it tells about the direction of the relationship, whether a factor is protective or increases the chance for having an outcome. So while interpreting data on logistic regression just saying that there was significant difference is not adequate, it is meaningful to state as to what extent having the factor increases or reduces the risk etc. kindly refer on how to properly interpret Odds ratios then re-write the results description.In table 2, you may need to indicate in the footnote as to which statistic was used to compute the *P-*values.Kindly check if there is a need for using chi-square statistic and computation of crude odds ratio, I would propose that you omit analysis with chi square, just start with computation of CoR then move to running the multivariate logistic regression using the same criteria (Variables with p-value of 0.2 or less in the univariate logistic regression).When filling values in the tables, once you have indicated (%) in the column heading, you don’t need to keep indicating % for each entry in the column. So remove all percentages in the cells and leave only that you have indicated in the column heading.

**Discussion**

I am of the opinion and I recommend the following structure of the discussion that could help to improve the paper. However, the authors can just ignore it.

1.  The authors have to start with: **Why did they do this study? ** (State in few sentences why it was important to conduct the reported study with reference with what is reported in the introduction/background section).

2.  **What exactly did you do**? (State in few sentence what you did)

3.  **What did you find?** (Summarize your main findings in 0.5 page/avoid repeating exactly what you stated in the results description in the results section)

4.  **Set the Main findings in context with other studies. **Discuss what is a) novel, b) similar and 3) different to other studies and what this means for control managers in connection to your study population

5.  **Show the limitations of the study** (discuss the limitation of the study design/compliance and hence the **Generalizability **of your findings) .

6. **What is the conclusion you can draw from your findings/results?** (Here you state the key message, carefully avoiding any speculations)

We look forward to receiving your revised manuscript.

Kind regards,

David Zadock Munisi, Ph.D

Academic Editor

PLOS ONE

Journal Requirements:

Reviewers' comments:

Reviewer's Responses to Questions

**Comments to the Author**

1. Is the manuscript technically sound, and do the data support the conclusions?

Reviewer #1: Partly

2. Has the statistical analysis been performed appropriately and rigorously? 

Reviewer #1: Yes

3. Have the authors made all data underlying the findings in their manuscript fully available?

Reviewer #1: Yes

4. Is the manuscript presented in an intelligible fashion and written in standard English?

Reviewer #1: Yes

5. Review Comments to the Author

Reviewer #1: Revision of the manuscript that takes into account the following points will improve its clarity:

1. The authors should adopt one single style for reference citation. There are, in addition, a number of references that completely lack the journal title, or it is included in the wrong place. These anomalies must be addressed.

2. With good reason, the authors have made reference to articles published in a collection in 2015 in the Korean Journal of Parasitology. Amongst those, several are of particularly high pertinence to their study, having been conducted in the same geographical area of Tanzania, although they refer to only one in the Introduction, preferring to cite other, rather less relevant studies. Curiously, furthermore, the article from that 2015 collection with perhaps the most relevance to their study - since it specifically addresses STH and schistosomiasis infections in adults (Prevalence of Schistosomes and Soil-Transmitted Helminths and Morbidity Associated with Schistosomiasis among Adult Population in Lake Victoria Basin, Tanzania. Siza, Kaatano et al) - is not cited at all. This anomaly, again, should be addressed, and the differences in their findings acknowledged and discussed. The reference to a study conducted elsewhere in Tanzania in an urban setting (Ref. #27, line 297) seems anomalous here.

3. It would be instructive to know what mass drug administration programmes were on-going - if any - in the study area prior to and during their study. The authors should include this information if available

4. Methods: an exhaustive description is not necessary, but the authors should include, at the very least, an overview of the Kato-Katz procedure used. Readers need to know if duplicate slides were made, if so; were they read independently by different microscopists, and, again if so, what procedure was implemented in the case of divergent results. Reference is made to a quality control assessment of 10% of samples by a 'second investigator', implying that only a single microscopist was initially responsible for diagnostic microscopy, which would be suboptimal.

5. Results: for the statistical comparisons presented in Tables 1 & 3, it is essential that the authors emphasize the fact that (especially for Table 3) the associations identified did not hold up following adjustment for confounders. It is equally important to provide a list of the variables included as confounders for the adjusted analyses. In addition, in the context of Table 3, the authors refer in the text to 'the habit of wearing shoes and the washing of hands after caring for infants', but neither of these variables appear in the analyses presented in Table 3, although they do appear subsequently in Table 5. This is confusing and should be addressed.

6. Discussion: line 299 erroneously gives the hookworm prevalence in their study as 4.3%.

7. Discussion lines 320-321: as mentioned above, in the context of 'interventions in place', the authors have not provided the relevant information concerning (pre-)existing interventions.

8. Discussion lines 344-367: this section begins with the statement 'In our study, all variables corresponding to the sanitation and hygiene factors were associated with STH infection, except the presence of a toilet in the household.' The same section then ends with the statement 'The logistic regression analysis didn’t confirm the association of STH infection with sanitation factors, and other factors likely play a major role in infection.' These two statements are completely contradictory and incompatible with each other. The results of the most appropriate statistical analyses (multivariate logistic regression with adjusted OR) revealed the absence of any associations, and it is therefore those analyses that should be the focus of discussion rather than appearing as almost a footnote at the end of the paragraph. As it stands, the predominant focus of the discussion is on the results of analyses that were not adjusted for confounders, which is erroneous. The authors should address this issue by modifying their discussion, including their opinion on what the 'other factors' they refer to might be.

9. Conclusions, line 398: here the authors refer to '....the improvement in WaSH', but it is not clear to what data or information they are referring. Since their study is cross-sectional in nature, any perceived improvement in WaSH must be based on an appropriate comparison of pre-existing (published) data with the data they generated here. This point must be clarified.

6. PLOS authors have the option to publish the peer review history of their article (what does this mean?). If published, this will include your full peer review and any attached files.

Reviewer #1: **Yes: **Adrian JF Luty

---

## [Author Response · Author response to Decision Letter 0]

14 Jun 2023

We thank the Editor and the Reviewer for providing comments to improve our manuscript. 

Please refer to the clean document, because some of the changes were made on the clear version of the document. 

Additionally, please find attach the data set supporting our findings. 

We addressed comments as follows. In blue there is the quotation of the corresponding comment. 

Introduction:

The authors mention STH without mentioning Strongyloides stercoralis as one important member of the STH group of helminths.

Thank you for emphasizing this point. We have added S. stercoralis in the introduction. 

Methods

Study settings

The authors have used 2012 population for Muleba while there is a new report with a different population. Kindly refer to the 2022 National Census report https://www.nbs.go.tz/nbs/takwimu/Census2022/Administrative_units_Population_Distribution_Report_Tanzania_volume1a.pdf

We have changed as needed. At the time of writing the submitted manuscript the Census was complete, but results were still not published in full. Thank you for underlining this point. 

Statistical analysis

Line 186: An independent sample t-test or χ2 was used to compare continuous and categorical variables. This statement is not clear, which one of the two was compared using t-test and which one with χ2 Please, make it clear.

This paragraph was rephrased as follows: Independent and dependent variables were summarized using descriptive statistics, which were reported as mean and standard deviation for continuous variables and frequencies and proportions for categorical ones. A χ2 was used to compare categorical variables

It is also worthwhile noting that, conducting the statistical tests intends at establishing whether there is an association between the independent and the dependent variable. The strength of logistic regression analyisis is that it tells about the direction of the relationship, whether a factor is protective or increases the chance for having an outcome. So while interpreting data on logistic regression just saying that there was significant difference is not adequate, it is meaningful to state as to what extent having the factor increases or reduces the risk etc. kindly refer on how to properly interpret Odds ratios then re-write the results description.

The result section has been entirely modified, and the statistical analysis was cross checked. Tables were modified accordingly. Please refer to the clear version of the manuscript. 

In table 2, you may need to indicate in the footnote as to which statistic was used to compute the P-values.

This was added. 

Kindly check if there is a need for using chi-square statistic and computation of crude odds ratio, I would propose that you omit analysis with chi square, just start with computation of CoR then move to running the multivariate logistic regression using the same criteria (Variables with p-value of 0.2 or less in the univariate logistic regression).

We have adjusted the report of cOR and a OR accordingly. 

When filling values in the tables, once you have indicated (%) in the column heading, you don’t need to keep indicating % for each entry in the column. So remove all percentages in the cells and leave only that you have indicated in the column heading.

The tables have been revised accordingly. 

Discussion

I am of the opinion and I recommend the following structure of the discussion that could help to improve the paper. However, the authors can just ignore it.

1. The authors have to start with: Why did they do this study? (State in few sentences why it was important to conduct the reported study with reference with what is reported in the introduction/background section).

2. What exactly did you do? (State in few sentence what you did)

3. What did you find? (Summarize your main findings in 0.5 page/avoid repeating exactly what you stated in the results description in the results section)

4. Set the Main findings in context with other studies. Discuss what is a) novel, b) similar and 3) different to other studies and what this means for control managers in connection to your study population

5. Show the limitations of the study (discuss the limitation of the study design/compliance and hence the Generalizability of your findings) .

6. What is the conclusion you can draw from your findings/results? (Here you state the key message, carefully avoiding any speculations)

We considered the new structure when building the revised draft. 

Comments from reviewer

We have addressed the comments of the Reviewer, that we thank, as follows. 

The authors should adopt one single style for reference citation. There are, in addition, a number of references that completely lack the journal title, or it is included in the wrong place. These anomalies must be addressed.

2. With good reason, the authors have made reference to articles published in a collection in 2015 in the Korean Journal of Parasitology. Amongst those, several are of particularly high pertinence to their study, having been conducted in the same geographical area of Tanzania, although they refer to only one in the Introduction, preferring to cite other, rather less relevant studies. Curiously, furthermore, the article from that 2015 collection with perhaps the most relevance to their study - since it specifically addresses STH and schistosomiasis infections in adults (Prevalence of Schistosomes and Soil-Transmitted Helminths and Morbidity Associated with Schistosomiasis among Adult Population in Lake Victoria Basin, Tanzania. Siza, Kaatano et al) - is not cited at all. This anomaly, again, should be addressed, and the differences in their findings acknowledged and discussed. The reference to a study conducted elsewhere in Tanzania in an urban setting (Ref. #27, line 297) seems anomalous here.

Thank you for emphasizing this point. We went through the corresponding section in the discussion and we used the suggested reference by Siza et al to comment on our findings. 

 It would be instructive to know what mass drug administration programmes were on-going - if any - in the study area prior to and during their study. The authors should include this information if available

We have inserted the following paragraph to explain the intervention measures in place in Muleba, in relation to the prevalence observed in the district “The 9.2% prevalence observed after seven years in our study, lower than what reported in the Lake Victoria setting previously suggests the effectiveness of interventions in place, including In Tanzania, annual MDA campaigns are conducted in primary schools as part of the national neglected tropical diseases (NTD) control programme” . 

4. Methods: an exhaustive description is not necessary, but the authors should include, at the very least, an overview of the Kato-Katz procedure used. Readers need to know if duplicate slides were made, if so; were they read independently by different microscopists, and, again if so, what procedure was implemented in the case of divergent results. Reference is made to a quality control assessment of 10% of samples by a 'second investigator', implying that only a single microscopist was initially responsible for diagnostic microscopy, which would be suboptimal.

We have added the lacking details in the method section. Because the procedure was sub-optimal, we included this consideration in the method section, by adding the following paragraph: “Additionally, we have collected and processed only a single stool sample from each participant, one only slide was prepared from each sample and analysed by one microscopist. Even though for quality checking 10 % of samples were cross-checked, this is sub-optimal, thus potentially underestimating the prevalence of STH among the studied population”.

5. Results: for the statistical comparisons presented in Tables 1 & 3, it is essential that the authors emphasize the fact that (especially for Table 3) the associations identified did not hold up following adjustment for confounders. It is equally important to provide a list of the variables included as confounders for the adjusted analyses. In addition, in the context of Table 3, the authors refer in the text to 'the habit of wearing shoes and the washing of hands after caring for infants', but neither of these variables appear in the analyses presented in Table 3, although they do appear subsequently in Table 5. This is confusing and should be addressed.

We have rephrased the corresponding paragraphs in the result section, to specify better that not all the difference in prevalence were confirmed by logistic regression. We kept the separate paragraph for the logistic regression to report the aOR. We have modified the heading of table 5, which referred to logistic regression and caused confusion (all tables contain the logistic regression column for the corresponding variables analysed) and the habit of wearing gumboots is included only in table 5. The revision of stats did not confirm the association of infection and caring for infants, we have changes all sections containing this information and all the tables, accordingly. 

6. Discussion: line 299 erroneously gives the hookworm prevalence in their study as 4.3%.

This was corrected to 7.43%, thank you for this. 

7. Discussion lines 320-321: as mentioned above, in the context of 'interventions in place', the authors have not provided the relevant information concerning (pre-)existing interventions.

We have added the reference to the MDA program as for national guidelines here, as done for the comment above. 

8. Discussion lines 344-367: this section begins with the statement 'In our study, all variables corresponding to the sanitation and hygiene factors were associated with STH infection, except the presence of a toilet in the household.' The same section then ends with the statement 'The logistic regression analysis didn’t confirm the association of STH infection with sanitation factors, and other factors likely play a major role in infection.' These two statements are completely contradictory and incompatible with each other. The results of the most appropriate statistical analyses (multivariate logistic regression with adjusted OR) revealed the absence of any associations, and it is therefore those analyses that should be the focus of discussion rather than appearing as almost a footnote at the end of the paragraph. As it stands, the predominant focus of the discussion is on the results of analyses that were not adjusted for confounders, which is erroneous. The authors should address this issue by modifying their discussion, including their opinion on what the 'other factors' they refer to might be.

We have rephrased the result section that was the primary cause of this issue. The discussion section has been changed according to the new structure of the manuscript. 

9. Conclusions, line 398: here the authors refer to '....the improvement in WaSH', but it is not clear to what data or information they are referring. Since their study is cross-sectional in nature, any perceived improvement in WaSH must be based on an appropriate comparison of pre-existing (published) data with the data they generated here. This point must be clarified.

We have rephrased conclusions, according to the cross-sectional nature of the study. 

Please refer to the clear version of the manuscript, because some of the additions were done there: the version with track changes contains major changes, but the final revision was made on the clear copy, because the one with track changes was confusing. 

Thank you again for the opportunity of improving the work, I hope , on behalf of all the co-authors, that we addressed the comments in a good way.

Dr Valeria Silvestri

---

## [Decision Letter · Decision Letter 1]

29 Jun 2023

PONE-D-23-02659R1Burden and factors associated with ongoing transmission of soil-transmitted helminths infections among the adult population: a community-based cross-sectional survey in Muleba district, TanzaniaPLOS ONE

Dear Dr. VALERIA,

Thank you for submitting your manuscript to PLOS ONE. After careful consideration, we feel that it has merit but does not fully meet PLOS ONE’s publication criteria as it currently stands. Therefore, we invite you to submit a revised version of the manuscript that addresses the points raised during the review process. Please submit your revised manuscript by Aug 13 2023 11:59PM. If you will need more time than this to complete your revisions, please reply to this message or contact the journal office at plosone@plos.org. Please include the following items when submitting your revised manuscript:A rebuttal letter that responds to each point raised by the academic editor and reviewer(s). You should upload this letter as a separate file labeled 'Response to Reviewers'.A marked-up copy of your manuscript that highlights changes made to the original version. You should upload this as a separate file labeled 'Revised Manuscript with Track Changes'.An unmarked version of your revised paper without tracked changes. You should upload this as a separate file labeled 'Manuscript'.If applicable, we recommend that you deposit your laboratory protocols in protocols.io to enhance the reproducibility of your results. Protocols.io assigns your protocol its own identifier (DOI) so that it can be cited independently in the future. For instructions see: https://journals.plos.org/plosone/s/submission-guidelines#loc-laboratory-protocols. Additionally, PLOS ONE offers an option for publishing peer-reviewed Lab Protocol articles, which describe protocols hosted on protocols.io. Read more information on sharing protocols at https://plos.org/protocols?utm_medium=editorial-email&utm_source=authorletters&utm_campaign=protocols.

We look forward to receiving your revised manuscript.

Kind regards,

David Zadock Munisi, Ph.D

Academic Editor

PLOS ONE

Journal Requirements:

Reviewers' comments:

Reviewer's Responses to Questions

**Comments to the Author**

1. If the authors have adequately addressed your comments raised in a previous round of review and you feel that this manuscript is now acceptable for publication, you may indicate that here to bypass the “Comments to the Author” section, enter your conflict of interest statement in the “Confidential to Editor” section, and submit your "Accept" recommendation.

Reviewer #1: (No Response)

2. Is the manuscript technically sound, and do the data support the conclusions?

Reviewer #1: Yes

3. Has the statistical analysis been performed appropriately and rigorously? 

Reviewer #1: Yes

4. Have the authors made all data underlying the findings in their manuscript fully available?

Reviewer #1: Yes

5. Is the manuscript presented in an intelligible fashion and written in standard English?

Reviewer #1: No

6. Review Comments to the Author

Reviewer #1: The revised manuscript is an improvement that addresses most of my comments.

I strongly suggest proof-reading and correction by a native English speaker to improve the clarity and comprehension.

7. PLOS authors have the option to publish the peer review history of their article (what does this mean?). If published, this will include your full peer review and any attached files.

Reviewer #1: **Yes: **Adrian JF Luty

---

## [Author Response · Author response to Decision Letter 1]

5 Jul 2023

Dear Editor, Dear Reviewers,

Thank you for the last comments to our work. Please find attached the revised version of the manuscript. 

We have checked references and fixed the issues detected. 

English carries at the same time the blessing and the burden of being a shared communication tool among Scientists in the world. Native speakers could write a perfect English version of our work, but the mindset behind certain language choices, words, length, expression, that reflects the cultural background and identity of authors from other parts of the world, would be lost. 

Of course, I hope we managed to fix grammar and major flaws. 

On behalf of the authors, I thank again and wish a good day. 

Dr. Valeria Silvestri

---

## [Editor Report · Decision Letter 2]

7 Jul 2023

Burden and factors associated with ongoing transmission of soil-transmitted helminths infections among the adult population: a community-based cross-sectional survey in Muleba district, Tanzania

PONE-D-23-02659R2

Dear Valeria,

We’re pleased to inform you that your manuscript has been judged scientifically suitable for publication and will be formally accepted for publication once it meets all outstanding technical requirements.

Kind regards,

David Zadock Munisi, Ph.D

Academic Editor

PLOS ONE

---

## [Editor Report · Acceptance letter]

18 Jul 2023

PONE-D-23-02659R2 

Burden and factors associated with ongoing transmission of soil-transmitted helminths infections among the adult population: a community-based cross-sectional survey in Muleba district, Tanzania 

Dear Dr. Silvestri:

I'm pleased to inform you that your manuscript has been deemed suitable for publication in PLOS ONE. Congratulations! Your manuscript is now with our production department. 

Kind regards, 

on behalf of

Dr. David Zadock Munisi 

Academic Editor

PLOS ONE